# Learning 1D Causal Visual Representation with De-focus Attention Networks

Chenxin Tao[1,3*], Xizhou Zhu[1,2*], Shiqian Su[1,3*], Lewei Lu[3], Changyao Tian[4], Xuan Luo[1], Gao Huang[1], Hongsheng Li[4], Yu Qiao[2], Jie Zhou[1], Jifeng Dai[1,2 ✉]

[1]Tsinghua University  [2]Shanghai Artificial Intelligence Laboratory
[3]SenseTime Research  [4]The Chinese University of Hong Kong
{tcx20,ssq20,luoxuan21}@mails.tsinghua.edu.cn,
{zhuxizhou,gaohuang,jzhou,daijifeng}@tsinghua.edu.cn,luotto@sensetime.com
tcyhost@link.cuhk.edu.hk, hsli@ee.cuhk.edu.hk, qiaoyu@pjlab.org.cn

## Abstract

Modality differences have led to the development of heterogeneous architectures for vision and language models. While images typically require 2D non-causal modeling, texts utilize 1D causal modeling. This distinction poses significant challenges in constructing unified multi-modal models. This paper explores the feasibility of representing images using 1D causal modeling. We identify an "over-focus" issue in existing 1D causal vision models, where attention overly concentrates on a small proportion of visual tokens. The issue of "over-focus" hinders the model's ability to extract diverse visual features and to receive effective gradients for optimization. To address this, we propose De-focus Attention Networks, which employ learnable bandpass filters to create varied attention patterns. During training, large and scheduled drop path rates, and an auxiliary loss on globally pooled features for global understanding tasks are introduced. These two strategies encourage the model to attend to a broader range of tokens and enhance network optimization. Extensive experiments validate the efficacy of our approach, demonstrating that 1D causal visual representation can perform comparably to 2D non-causal representation in tasks such as global perception, dense prediction, and multi-modal understanding. Code shall be released.

## 1 Introduction

Due to inherent modality differences, vision and language models have evolved into distinct heterogeneous architectures. A key difference is that images usually require 2D non-causal modeling, while texts often utilize 1D causal modeling. This distinction presents a significant challenge in constructing unified multi-modal models. Many existing multi-modal models [37, 3, 11, 5, 29] have to train vision and language encoders separately before combining them. A crucial question in advancing unified vision-language modeling is how to represent images using 1D causal modeling.

Following the success of causal language modeling (e.g., GPT-series [52, 53, 8]), some studies [10, 17] have explored causal modeling in the vision domain. These efforts primarily focus on auto-regressive visual pre-training by adding a causal attention mask to standard Transformers [15]. Despite numerous attempts, the gap between 1D causal and 2D non-causal vision models remains unbridged. As shown in Sec. 5, many 1D causal vision models, such as State Space Models [61, 21] and causal ViTs [15], perform inferiorly compared to their modified 2D non-causal counterparts.

---

*Equal contribution. This work is done when Chenxin Tao and Shiqian Su are interns at SenseTime Research.
✉ Corresponding to Jifeng Dai <daijifeng@tsinghua.edu.cn>.

38th Conference on Neural Information Processing Systems (NeurIPS 2024).

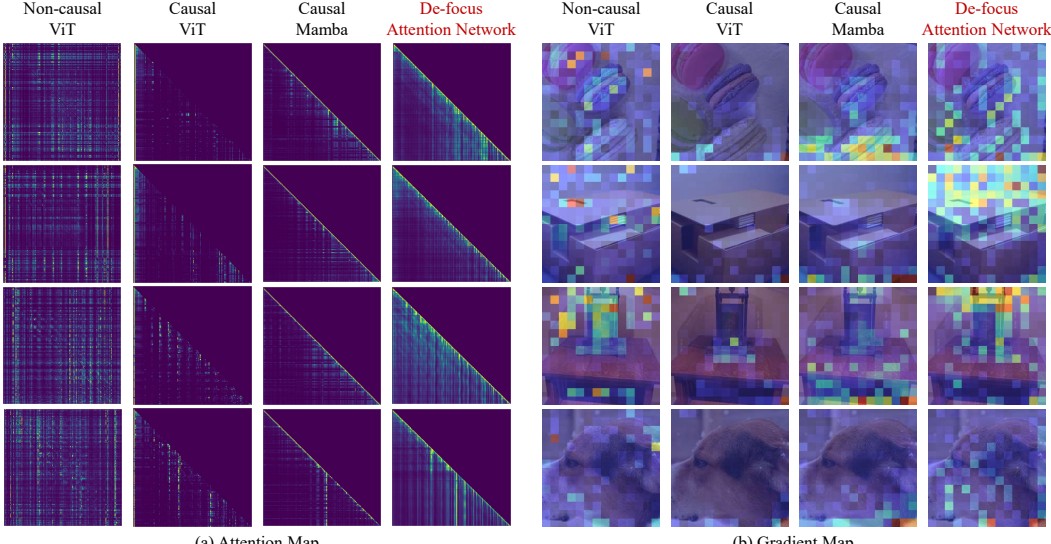

| Non-causal ViT | Causal ViT | Causal Mamba | De-focus Attention Network | Non-causal ViT | Causal ViT | Causal Mamba | De-focus Attention Network |

(a) Attention Map  (b) Gradient Map

Figure 1: **Visualizations of (a) Attention Map and (b) Gradient Map of different models**, including Non-causal ViT, Causal ViT, Causal Mamba and our De-focus Attention Network (Mamba-based). The results are from the 11[th] layer of ViT (12 in total) and 22[nd] layer of Mamba (24 in total). (a) The approximated attention maps of all image tokens: The row and column axes represent the query and key token index respectively. Brighter color indicates larger attention values. (b) The gradient maps of each image token input after back-propagation: Redder colors indicate larger gradient norms. See Appendix A for more visualizations on different layers.

In this paper, we identify an "over-focus" issue in existing 1D causal vision models. Fig. 1 visualizes the attention and gradient maps of several ImageNet-trained networks, including 2D non-causal ViT, 1D causal ViT, and 1D causal Mamba. The results show that in 1D causal vision models, the attention patterns are overly concentrated on a small proportion of visual tokens, especially in the deeper network layers close to the output. This phenomenon hinders the model from extracting diverse visual features during the forward calculation and obtaining effective gradients during the backward propagation. We refer to this phenomenon as the "over-focus" issue in 1D causal vision models.

To address the issue, a "de-focus attention" strategy is introduced. The core idea is to guide the network to attend to a broader range of tokens. First, learnable bandpass filters are introduced to filter different sets of token information, and then combine their attention patterns. This ensures that even if over-focusing occurs, the attention pattern remains diverse due to the varying constraints of each set. Second, optimization strategies are improved. A large drop path rate is employed to encourage the network to attend to more tokens within one layer, rather than relying on depth to get large receptive fields. For tasks requiring global understanding (e.g., image classification), an auxiliary loss is applied to the globally pooled features to enhance the effective gradients for all tokens in the sequence.

Extensive experiments demonstrate the effectiveness of our De-focus Attention Networks for 1D causal visual representation learning. It achieves comparable or even superior performance to 2D non-causal ViTs across various tasks, including image classification, object detection, and image-text retrieval. Our method has been validated on both ViTs and Mambas. Our contributions can be summarized as follows:

- We identify the over-focus issue in 1D causal visual modeling, where the model overly focuses on a small proportion of visual tokens in the deeper layers of the network.

- To address this issue, we propose a "de-focus attention" strategy. This involves integrating learnable bandpass filters into the existing attention operators to achieve diverse attention patterns. Additionally, we introduce a large drop path probability and an auxiliary loss on average pooled features during training to enhance network optimization.

- Our De-focus Attention Networks have demonstrated that 1D causal visual representation can achieve performance equivalent to 2D non-causal representation in tasks requiring global perception, dense prediction and multi-modal understanding tasks.

## 2 Related Work

**State Space Models (SSMs)** are intrinsically causal models, originated from the classic Kalman filter[31]. SSMs describe the behavior of continuous-dynamic systems, enabling parallel training and linear complexity inference. [24] proposed a Linear State Space Layer, merging the strengths of continuous-time models, RNNs and CNNs. HIPPO [22] introduced methods to facilitate continuous-time online memorization. Building on these foundations, Structured SSMs (e.g., S4 [23], Diagonal State Spaces (DSS) [25], S5 [58]), Recurrent SSMs (e.g., RWKV [49], LRU [45]) and Gated SSMs (e.g., GSS[42], Mega[41]) further expand the SSMs landscape. Notably, Mamba [21] excels in long-sequence modeling with its selective scan operator for information filtering and hardware-aware algorithms for efficient storage of intermediate results. As SSMs have drawn more and more attention recently, they also have extensive applications in domains that need long sequences processing such as medical [44, 6], video [34], tabular domain [2] and audio/speech [20, 30]. These successes achieved by SSMs prompt us to explore their application in visual modeling within this causal framework.

**2D Non-Causal Visual Modeling** are dominant in vision domains. Convolutional Neural Networks (CNNs), operating in a 2D sliding-window manner [33] with inductive biases such as translation equivariance and locality, have demonstrated remarkable adaptability [32, 57, 62, 71, 28, 27, 63]. Vision Transformers (ViTs) [15] utilize a non-causal self-attention mechanism, enabling global receptive fields. Subsequent improvements focus on enhancing locality[39], refining self-attention mechanisms[74, 4], and introducing novel architectural designs [69, 70, 46, 26], while maintaining non-causality. Recent advances in State Space Models (SSMs) have inspired new vision backbone networks, such as VMamba [38], Vision Mamba [75], and Vision-RWKV [16]. Although SSMs are inherently causal, these works incorporate non-causal adjustments to enhance vision performance. VMamba introduced a four-way scanning strategy, Vision Mamba incorporated bidirectional SSMs, and Vision-RWKV adopted bidirectional global attention and a special token shift method. These designs of arrangement hinder the unification of vision and language modeling.

**1D Causal Visual Modeling.** While 1D causal modeling has primarily been used in language[7] and speech[66], it has also been explored for visual representation. In recent years, the causal visual modeling has been adopted in Transformer-based visual generation methods such as Image Transformer [47] and VQGAN [18]. These models first discretize images into grids of 2D tokens, which are then flattened for auto-regressive learning. However, their performance significantly lags behind [48, 1]. Of particular interest, iGPT [10] also employed auto-regressive causal modeling for pre-training, followed by linear probing or fine-tuning to achieve commendable results in various downstream tasks, though still worse than non-causal models [14, 9]. Similarly, AIM [17] applied causal masks to the self-attention layers, and pre-trained with an auto-regressive objective, showing good scaling potential. Despite many attempts, the performance gap between 1D causal and 2D non-causal vision models remains.

## 3 Preliminary

**Transformers** [67] with causal attention consist of multiple attention layers. Each attention layer computes a weighted average feature from the preceding context for every input token, with aggregated features weighted by the similarities between tokens. The attention layer is written as:

$$y_t = \sum_{s \leq t} \text{Softmax}(Q_t^\top K_s) V_s, \tag{1}$$

where $s$ and $t$ are indexes of different locations of the input sequence, $Q_i, K_i, V_i$ are projections of input $x_i$, and $y_t$ is the output of the attention layer.

**State Space Models (SSMs)** are classical latent state models widely used in various scientific fields [44, 6, 34, 68, 50, 73]. Originally, SSMs are defined for continuous signals, mapping a 1D input signal $x(t) \in \mathbb{R}$ to a latent state $h(t) \in \mathbb{R}^N$ and computing the output $y(t) \in \mathbb{R}$ from the latent state. To apply SSMs to discrete sequences, their discrete form is defined as

$$h_t = A_t h_{t-1} + K_t x_t, \qquad y_t = Q_t^\top h_t, \tag{2}$$

where $A_t \in \mathbb{R}^{N \times N}$, $K_t \in \mathbb{R}^{N \times 1}$, $Q_t \in \mathbb{R}^{N \times 1}$ are parameters of the system. Note that we use notations different from the original SSMs ($K_t, Q_t$ instead of $B_t, C_t$) for a better comparison with Transformers above.

SSMs can also be transformed into another formulation by expanding the recurrent process:

$$y_t = \sum_{s \leq t} Q_t^\top \left( A_t \ldots A_{s+1} \right) K_s x_s. \tag{3}$$

This formulation resembles the conventional attention module and explicitly reveals the relationship between different inputs in the sequence. We use this form for further discussion.

There are multiple variants of SSMs, mainly differing in the parameterization of $(A_t, K_t, Q_t)$. We introduce some well-known SSMs and discuss their differences below.

*RetNet* [61] and *Transnormer* [51] employ a fixed $A$ and convert it into an exponential decay (defined by $\lambda \in \mathbb{R}$) with a relative positional embedding (defined by $\theta \in \mathbb{R}^N$):

$$y_t = \sum_{s \leq t} Q_t^\top \underbrace{e^{\lambda(t-s)}}_{\text{exp decay}} \underbrace{e^{i\theta(t-s)}}_{\text{relative pos embed}} K_s x_s. \tag{4}$$

*Mamba* [21] and *S4* [23] use zero-order hold (ZOH) rule for discretization, introducing a time-scale parameter $\Delta_t$. The discretization rule is $A_t = \exp(\Delta_t \hat{A})$ and $K_t = (\Delta_t \hat{A})^{-1}(\exp(\Delta_t \hat{A}) - I) \cdot \Delta_t \hat{K}_t$, where $\hat{A}$ and $\hat{K}_t$ are learnable parameters. S4 uses data-independent parameters, while Mamba computes these parameters based on inputs. The formulation can be written as:

$$y_t = \sum_{s \leq t} Q_t^\top \underbrace{\exp\big(\hat{A}(\Delta_{s+1} + \cdots + \Delta_t)\big)}_{\text{learnable exponential decay}} K_s x_s. \tag{5}$$

## 4 Method

This section introduces our De-focus Attention Networks for 1D causal visual representation learning. Sec. 4.1 elucidates the main components of De-focus Attention as Learnable Bandpass Filter, while Sec. 4.2 further discusses two training strategies adopted in De-focus Network. The overall architecture of our model is presented in Fig. 2.

### 4.1 De-focus Attention with Learnable Bandpass Filter

To de-focus on a few salient tokens and enhance the extraction of diverse features from images, learnable bandpass filters are incorporated to first adaptively filter diverse information from the input and their attention patterns are then combined together. Due to the varying contents from different filters, the attention can still be diverse even if the over-focus issue happens.

These bandpass filters can be implemented through exponential spatial decay and relative position embedding similar to those in RoPE [59] and xPos [60], both of which are further made learnable. Our results demonstrate that these factors are crucial for the model to learn diverse attention patterns.

To show how spatial decay and relative position embeddings work as a bandpass filter, consider a simplified version of 1D causal attention equipped with them:

$$y(t) = \int_{s \leq t} e^{\lambda(t-s)} e^{i\theta(t-s)} x(s) \mathrm{d}s, \tag{6}$$

where $x(s)$ is the input signal at time $s$. $e^{\lambda(t-s)}$ ($\lambda < 0$) represents the simplest version of exponential spatial decay, which is also used by RetNet [61] and Transnormer [51]. $e^{i\theta(t-s)}$ is the relative position embedding proposed by RoPE [59] and xPos [60]. Here, the continuous time domain is used to facilitate derivation without losing generality.

The above equation implies a time domain convolution between $e^{\lambda(t-s)} e^{i\theta(t-s)}$ and $x(s)$. By transforming Eq. (6) into the frequency domain and using $\hat{x}(\omega), \hat{y}(\omega)$ to represent Fourier transform of corresponding $x(s), y(t)$, the frequency domain expression becomes:

$$\hat{y}(\omega) = \frac{1}{-\lambda + i(\omega - \theta)} \hat{x}(\omega), \qquad \|\hat{y}(\omega)\| = \frac{1}{|\lambda|} \frac{1}{\sqrt{1 + (\frac{\omega - \theta}{\lambda})^2}} \|\hat{x}(\omega)\|. \tag{7}$$

This equation indicates that Eq. (6) is actually a bandpass filter, where $\theta$ is its center frequency and $\lambda$ controls its passband width. Eq. (7) presents some interesting properties of 1D causal modeling:

1. If there is no spatial decay or relative position embedding (e.g., Transformers without $\mathrm{Softmax}$), Eq. (6) will degenerate to a summation of the inputs, losing the ability to filter spatial information;

2. If there is no relative position embedding (e.g., Mamba), 1D causal attention will perform low-pass frequency filtering, causing the query to miss the full information of features and resulting in information loss;

3. If only relative position embedding is used, it will degenerate to specific frequency selecting, which may also result in information loss;

4. If both spatial decay and relative position embedding are used (suggested), 1D causal attention will act as a bandpass filter. For a given query, when different components of the feature vector use different center frequencies (i.e., different $\theta$) and passbands width (i.e., different $\lambda$), a more diverse range of information will be gathered. Due to the diverse frequency passbands, even if the over-focus issue occurs, the attention remains diverse across different components of the feature vector.

To fully leverage the bandpass filtering mechanism, a learnable one is preferable. Experiments demonstrate that performance worsens when values are fixed or not well set.

Our De-focus Attention can be incorporated into different architectures. Below, examples of its implementation in causal ViT and Mamba are presented.

**De-focus Causal ViT.** ViT has additional attention activation (i.e., $\mathrm{Softmax}$) compared with SSMs. Learnable exponential spatial decay and learnable relative position embeddings are appended before applying the attention activation, following the common implementation of RoPE, as shown below:

$$y_t = \sum_{s \leq t} \mathrm{Softmax}\big(Q_t^\top \underbrace{e^{\lambda(t-s)}}_{\text{learnable decay}} \underbrace{e^{i\theta(t-s)}}_{\text{learnable RoPE}} K_s\big)x_s, \qquad (8)$$

where the terms of $e^\lambda$ and $e^{i\theta}$ function as the learnable bandpass filter.

**De-focus Mamba.** Since Mamba already has learnable and data-dependent exponential spatial decay, only attachment of learnable relative position embeddings to it is necessary:

$$y_t = \sum_{s \leq t} Q_t^\top \underbrace{\exp\big(\hat{A}(\Delta_{s+1} + \cdots + \Delta_t)\big)}_{\text{learnable exponential decay}} \underbrace{e^{i\theta(t-s)}}_{\text{learnable RoPE}} K_s x_s, \qquad (9)$$

where the terms of $\hat{A}\Delta$ and $e^{i\theta}$ function as the learnable bandpass filter.

### 4.2 De-focus Attention in Network Optimization

During network training, performance of 1D causal models can be further enhanced with improved optimization strategies. Specifically, using a large drop path rate with a linear schedule helps the model attend to more tokens in each layer. Additionally, applying an auxiliary loss to the global average feature mitigates the under-learning of features in deeper layers. The effects of these training strategies are illustrated in Fig. 3.

**Large Drop Path Rate with Linear Schedule.** Two ways for the final prediction to access information from previous inputs are observed: 1) *Network Depth*: Progressively looking forward a few tokens in each layer until reaching the earliest tokens; 2) *Intra-Layer Attention*: Using the attention mechanism within the same layer to directly capture information from more distant tokens.

Our goal is for each layer to fully utilize the existing attention mechanism to capture more and further information in one layer. Therefore, a large drop path rate (up to 0.7) is employed to encourage the network to rely less on depth and rely more on training the attention mechanism in each layer. Since a large drop path rate may hinder the model when only a few features are learned, i.e., at the beginning of training, a linear schedule that gradually increase the drop path rate is followed.

Fig. 3 demonstrates the effectiveness of this strategy, indicating that without large drop path strategies, the network tends to prefer to see less tokens in one layer and rely on network depth to increase the receptive field.

**Auxiliary Loss for Image Classification.** To address over-focus issue in backward gradients, an auxiliary loss is proposed to enrich the gradients variety and aid in the representation learning of

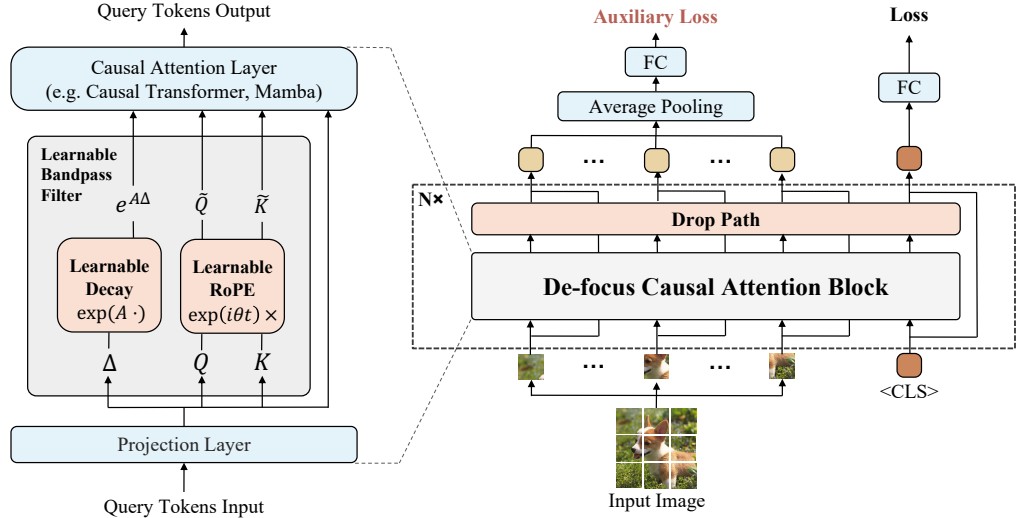

Figure 2: **Architecture of our De-focus Attention Network.** *Left:* Detailed architecture of De-focus Attention Block: The input tokens are projected to $Q, K$, and other parameters required by certain causal attention layer (e.g. Transformer or Mamba). $\Delta$ is data-dependent in De-focus Mamba, while is set to 1 in De-focus ViT. Learnable decay and learnable relative position embeddings form a learnable bandpass filter and are calculated before being fed into the causal attention layer. Parameter $\lambda$ in De-focus ViT corresponds to $A$ in this figure. *Right:* Overall architecture of De-focus Attention Network: Drop paths are incorporated after each De-focus Attention Block. All output image tokens are passed through Average Pooling and a fully connected layer to produce the auxiliary loss.

unnoticed tokens. The final representations of all image tokens (excluding the final <CLS> token) are averaged and fed into an additional linear layer. The auxiliary loss function is defined as the cross-entropy loss between the output of the additional linear layer and the ground truth label. This approach helps enrich the backpropagated gradients, thereby addressing the over-focus issue.

As shown in Fig. 3, after applying the auxiliary loss strategy, the backward gradients are significantly improved in its density, globality, and diversity in deeper layers.

## 4.3 Overall Architecture.

The overall architecture of our De-focus Attention Networks is illustrated in Fig. 2. The following explains how learnable bandpass filters and optimization strategies are integrated into existing models.

**De-focus Attention Blocks.** Each block consists of three main parts, which are a projection layer, a learnable bandpass filter, and a causal attention layer. The input tokens are first projected into a query $Q$, a key $K$. $\Delta$ is data-dependent in De-focus Mamba, while is set to 1 in De-focus ViT. Other projections may be required by the causal attention layer. The block has learnable decay parameters $A$ (corresponds to $\lambda$ in De-focus ViT) and learnable relative position embedding parameters $\theta$. Given these learnable parameters, exponential spatial decay and relative position embedding are computed as illustrated in Eq. (8) and Eq. (9). $Q, K$ and the exponential spatial decay term are integrated into the causal attention layer. Thus, the outputs of a De-focus Attention block aggregate the input information filtered by a series of learnable bandpass filters.

**De-focus Attention Networks.** Given an image, our De-focus Attention Networks first transform it into a sequence of image tokens and append an extra <CLS> token to the sequence end. The whole network then stacks $N$ De-focus Attention blocks to process the input sequence. Each block is equipped a drop path rate, which increases linearly during training. In the final layer, the <CLS> token is fed through a linear layer and used to compute a cross-entropy loss with class labels. All image tokens, excluding the final <CLS> token, are averaged and passed through a separate linear layer. An auxiliary cross-entropy loss is applied to this projected averaged feature. The two losses are then added with equal weights to form the final loss function.

Table 1: Comparison of causal and non-causal attentions for image classification on ImageNet-1K.

| Method | Causal | Size | #Param | ImageNet Top-1 Acc |
|---|---|---|---|---|
| DeiT-Small [64] | | $224^2$ | 22.1M | 79.9 |
| Mamba-ND-Small [35] | | $224^2$ | 24M | 79.4 |
| Vision Mamba-Small [75] | | $224^2$ | 26M | **80.5** |
| Vision RWKV-Small [16] | | $224^2$ | 23.8M | 80.1 |
| DeiT-Small | ✓ | $224^2$ | 22.1M | 78.6 |
| Mamba-Small [21] | ✓ | $224^2$ | 24.7M | 78.7 |
| Mamba-ND-Small [35] | ✓ | $224^2$ | 24M | 76.4 |
| De-focus ViT-Small | ✓ | $224^2$ | 22.4M | 79.6 |
| De-focus Mamba-Small | ✓ | $224^2$ | 25.8M | **80.3** |
| DeiT-Base [64] | | $224^2$ | 86.6M | **81.8** |
| S4ND-ViT-B [44] | | $224^2$ | 88.8M | 80.4 |
| Vision RWKV-Base [16] | | $224^2$ | 93.7M | **82.0** |
| De-focus ViT-Base | | $224^2$ | 87.4M | **81.8** |
| DeiT-Base | ✓ | $224^2$ | 86.6M | 80.1 |
| RetNet-Base [61] | ✓ | $224^2$ | 93.6M | 79.0 |
| Mamba-Base [21] | ✓ | $224^2$ | 91.9M | 80.5 |
| De-focus ViT-Base | ✓ | $224^2$ | 87.4M | 81.5 |
| De-focus RetNet-Base | ✓ | $224^2$ | 94.1M | 81.7 |
| De-focus Mamba-Base | ✓ | $224^2$ | 94.1M | **82.0** |
| ViT-Large [15] | | $384^2$ | 309.5M | 85.2 |
| Vision RWKV-Large [16] | | $384^2$ | 334.9M | **86.0** |
| De-focus Mamba-Large | ✓ | $384^2$ | 327.4M | 85.4 |

# 5 Experiments

## 5.1 Experiment Setup

**Implementation Details**. The De-focus Attention mechanisms are integrated into Mamba, RetNet, and ViT, referred to as De-focus Mamba, De-focus RetNet, and De-focus ViT, respectively. To improve optimization stability, $\lambda = -\exp(\hat{\lambda})$ is used and $\hat{\lambda}$ is the parameter to be optimized. In De-focus ViT and De-focus RetNet, different $\lambda$s are assigned to different heads. Mamba inherently implements data-dependent decay $\hat{A}\Delta$, where $\hat{A}$ is a learnable parameter and $\Delta$ is a projection from the input. The drop path rate increases following a linear schedule from 0.1 to 0.7.

**Image Classification.** ImageNet-1k [13] is used, which contains 1.28M images for training and 50K images for validation. The training recipe of DeiT [64] is followed. The small- and base-size models are trained on ImageNet for 300 epochs. The large-size model is firstly pre-trained on ImageNet-21k [55] for 90 epochs, and then fine-tuned on ImageNet-1k for 20 epochs. The AdamW optimizer [40] with a peak learning rate of 5e-4, a total batch size of 1024, a momentum of 0.9, and a weight decay of 0.05 are used. These models are trained on 32 Nvidia 80G A100 GPUs for 30 hours.

**Object Detection.** The MS-COCO dataset [36] and the DINO detection framework [72] are used, with different networks serving as the backbones. The De-focus Attention Networks implemented here are pre-trained on ImageNet-1K dataset for 300 epochs. These models are trained on 16 Nvidia 80G A100 GPUs for 20 hours.

The entire network is fine-tuned using both a 1× schedule (12 epochs) and a 3× schedule (36 epochs). The base learning rate is set to 2e-4, with a multi-step learning rate strategy employed to decrease it by a factor of ten after 11 epochs (1× schedule) or after 27 and 33 epochs (3× schedule). The weight decay and the total batch size is set to 1e-4 and 16, respectively.

**Contrastive Language-Image Pre-training (CLIP).** The Laion-400M dataset [56] is used for pre-training. Strategy introduced in OpenCLIP [12] is followed to train the model for 32 epochs. The zero-shot classification performance is evaluated on ImageNet-1K. The AdamW optimizer [40] is employed with a peak learning rate of 5e-4, a total batch size of 32768, a momentum of 0.9, and a weight decay of 0.1. These models are trained on 128 Nvidia 80G A100 GPUs for 128 hours.

Table 2: Results of object detection on the COCO [36] dataset with DINO [72] detector.

| Method | Causal | #Param | Epochs | $AP^{box}$ | $AP^{box}_{50}$ | $AP^{box}_{75}$ |
|---|---|---|---|---|---|---|
| ResNet-50[72] | | 47M | 12 | 49.0 | 66.6 | 53.5 |
| DeiT-Base | | 110M | 12 | 49.1 | **69.9** | 52.7 |
| De-focus ViT-Base | ✓ | 113M | 12 | 48.9 | 67.1 | 53.3 |
| De-focus Mamba-Base | ✓ | 115M | 12 | **50.8** | 68.9 | **55.2** |
| ResNet-50[72] | | 47M | 36 | 50.9 | 69.0 | 55.3 |
| DeiT-Base | | 110M | 36 | 52.3 | **72.5** | 56.7 |
| De-focus Mamba-Base | ✓ | 115M | 36 | **53.5** | 71.9 | **58.3** |

## 5.2 Main Results

**Image Classification**. The classification results are presented in Table 1. Evaluation of different types of De-focus Networks at various scales is conducted, with comparisons to both causal and non-causal models. The results show that previous causal models have inferior performance. In contrast, our model defies this trend, significantly outperforming other 1D causal models and achieving comparable performance to 2D non-causal models.

Notably, the De-focus Attention mechanism works well across various networks, e.g., Causal ViT, Mamba, and RetNet. And as the model size increases from small to large, it remains on par with the 2D non-causal ViTs.

**Object Detection**. As shown in Table 2, De-focus Mamba remarkably outperforms non-causal models such as DeiT and ResNet-50. This trend of superior performance persists even with an increasing number of training epochs. Additionally, excellent performance on the $AP^{box}_{75}$ metric may suggest that De-focus Attention Networks are more effective at fine-grained localization.

Table 3: Results on zero-shot image classification of CLIP pre-trained models.

| Method | Causal | #Param | ImageNet Zero-shot Top-1 Acc |
|---|---|---|---|
| OpenAI CLIP-Base/32 [54] | | 151.3M | 63.3 |
| OpenCLIP-Base/32 [12] | | 151.3M | 62.9 |
| De-focus Mamba-Base/32 | ✓ | 161.9M | 62.7 |

Table 4: Results on image-text retrieval on the COCO [36] dataset of CLIP pre-trained models.

| Method | Causal | #Param | Image Retrieval | | | Text Retrieval | | |
|---|---|---|---|---|---|---|---|---|
| | | | Recall@1 | Recall@5 | Recall@10 | Recall@1 | Recall@5 | Recall@10 |
| OpenAI CLIP-Base/32 [54] | | 151.3M | 30.4 | 55.0 | 65.7 | 49.2 | 73.4 | 82.4 |
| OpenCLIP-Base/32 [12] | | 151.3M | 35.3 | 61.0 | 71.8 | 52.5 | 77.0 | 84.9 |
| De-focus Mamba-Base/32 | ✓ | 161.9M | 34.6 | 60.3 | 71.2 | 51.7 | 76.3 | 84.8 |

**Image-text CLIP Pre-training**. The model is pre-trained using OpenCLIP to demonstrate its outstanding performance on large-scale image-text training. As shown in Table 3, the model performs comparably to 2D non-causal models. We also report cross-modal retrieval results in Table 4, which further validate that the learned causal feature can achieve similar results with non-causal features. These results indicate that the model has a similar scaling law to non-causal ViTs on larger dataset, demonstrating its robustness and scalability across various of tasks and datasets. Additionally, this experiment demonstrates the potential of 1D causal modeling for unified vision-language modeling.

## 5.3 Ablation Study

**Learnable Bandpass Filter.** As discussed in Sec. 4.1, exponential spatial decay and relative position embedding (RoPE) together act as a bandpass filter. Tab. 5(a) shows the effects of different configurations. When decay is not used, the performance significantly deteriorates. Employing learnable decay leads to an improvement of approximately 0.5% compared to fixed decay, while learnable RoPE can further enhance performance by 0.8%. In contrast, the data-dependent decay used in Mamba only results in a marginal improvement of 0.1%. These results indicate the integration of learnable decay and RoPE are necessary for good performance.

Table 5: Ablation studies of various design choices of De-focus Mamba-Base model on ImageNet-1k [13]. The default settings are set as (a) dpr = 0.4, with auxiliary loss, (b) with auxiliary loss, data dependent decay and learnable RoPE, (c) dpr = 0.4, with data dependent decay and learnable RoPE. "dpr" is drop path rate. The text in (c) denotes the input feature for the loss function.

(a) Ablation on Bandpass Filter

| Decay | RoPE | Acc |
|---|---|---|
| w/o | w/o | 75.2 |
| w/o | fixed | 75.3 |
| fixed | w/o | 79.9 |
| fixed | fixed | 80.0 |
| fixed | learnable | 80.6 |
| learnable | w/o | 80.4 |
| learnable | learnable | 81.2 |
| data dependent | learnable | 81.3 |

(b) Ablation on Drop Path.

| Drop Path | Acc |
|---|---|
| 0.1 | 79.6 |
| 0.4 | 81.6 |
| 0.7 | 80.9 |
| linear(0.1, 0.7) | 82.0 |

(c) Ablation on Loss Function

| Loss | Aux Loss | Acc |
|---|---|---|
| `<CLS>` | – | 81.6 |
| avg | – | 77.2 |
| `<CLS>` + avg | – | 79.7 |
| `<CLS>` | avg | 82.0 |

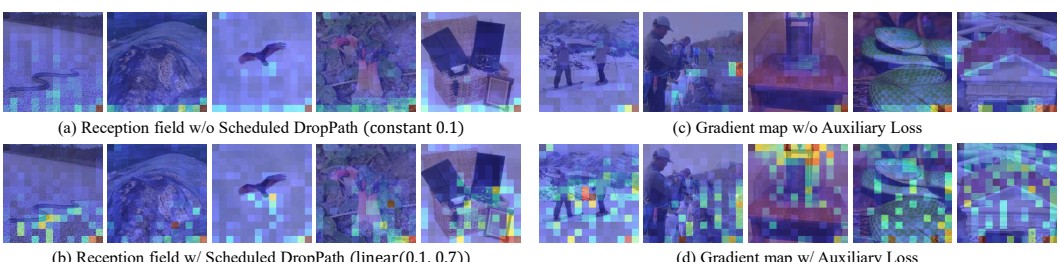

(a) Reception field w/o Scheduled DropPath (constant 0.1)  (c) Gradient map w/o Auxiliary Loss

(b) Reception field w/ Scheduled DropPath (linear(0.1, 0.7))  (d) Gradient map w/ Auxiliary Loss

Figure 3: **Qualitative ablation results of using scheduled drop path and auxiliary loss.** *(a)-(b)*: The receptive fields of our model trained with and without scheduled drop path. The scheduled drop path strategy enables a larger receptive field, facilitating the capture of denser semantic details. *(c)-(d)*: The backward gradient maps of our model trained with and without auxiliary loss. When trained with the auxiliary loss, the model can attend to denser and more diverse image tokens, particularly those at the front of the sequence.

**Drop Path.** Tab. 5(b) shows the performance of different drop path strategies, with rates ranging from 0.1 to 0.7. The best performance is achieved with a scheduled drop path rate $\mathrm{linear}(0.1, 0.7)$. Fig. 3(a)-(b) visualize the receptive field of the 22$^{\mathrm{nd}}$ layer of the network. The results demonstrate that using a large and scheduled drop path rate strategy allows for larger receptive field and helps capture more dense semantic details.

**Auxiliary Loss.** Tab. 5(c) compares various implementations of the loss function, which are generated from `<CLS>` token only, average token only, concatenation of `<CLS>` token and average token, and `<CLS>` token with auxiliary average token. The results reveal that the average pooled feature alone performs poorly in training the network. It may result from the fact that previous tokens often have incomplete information. However, it serves as an effective auxiliary component, thereby enhancing the network training. The visualization of gradient maps at the 22$^{\mathrm{nd}}$ layer of the network are shown in Fig. 3(c)-(d). When training with auxiliary loss, the density, globality, and diversity of backward gradients are significantly improved.

## 6 Conclusion

We propose De-focus Attention Networks to enhance the performance of causal vision models by addressing the issue of over-focus in them. The over-focus phenomenon, i.e. attention pattern is overly focused on a small proportion of visual tokens, is observed both during the forward calculation and backpropagation. These De-focus models incorporate a decay mechanism and relative position embeddings, functioning together as diverse and learnable bandpass filters to introduce various attention patterns. The models are trained with a large scheduled drop path rate and auxiliary loss to enhance the density, globality, and diversity of backward gradients. A series of De-focus models based on Mamba, RetNet, and ViT significantly outperform other causal models and achieve comparable or even superior performance to state-of-the-art non-causal models. By implementing the de-focus

strategy, our work bridges the performance gap between causal and non-causal vision models, paving the way for the development of state-of-the-art unified vision-language models.

**Limitations.** While the De-focus Attention has achieved very promising results, some differences between causal and non-causal vision models still need further exploration. For example, compared with non-causal models, De-focus Mamba-Base excels at $AP_{75}^{box}$ metric but is inferior at $AP_{50}^{box}$ metric on object detection. It is worth studying how causal models perform dense prediction tasks. Furthermore, currently we only explore 1D causal modeling in purely visual settings, while exploring unified 1D causal modeling of vision and language remains to be verified.

**Broader Impacts.** De-focus Attention Networks demonstrate the potential of building unified multi-modal models, so they may also cause similar problems as previous works. For instance, it may also require huge computational resources, contain dataset bias, and raise ethical concerns when being adopted for training large multi-modal foundation models.

**Acknowledgements.** The work is partially supported by the National Natural Science Foundation of China under Grants 62321005.

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

# A    More Experiment Results

## A.1    Visualization

This section provides visualization results of attention maps and gradient maps from more different layers of different models, as shown in Fig. 4, Fig. 5 and Fig. 6. Compared to other causal models, our de-focus attention network has denser attention maps and diverse gradient maps across all layers.

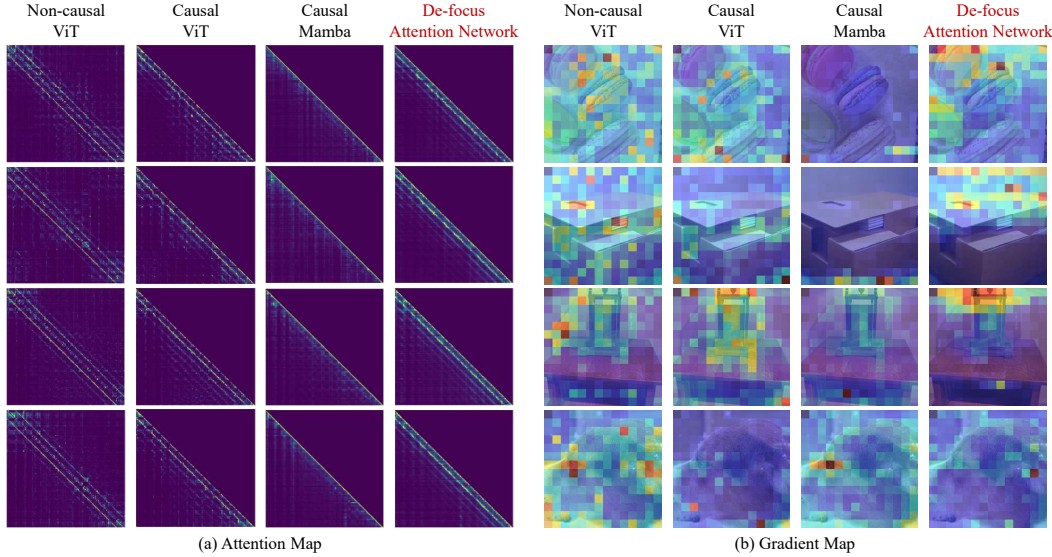

Figure 4: **Visualizations of Attention Map (a) and Gradient Map (b) of different models**, including non-causal ViT, causal ViT, Causal Mamba and our De-focus Attention Network (Mamba-based). The results are from the $3^{rd}$ layer of ViT (12 in total) and $6^{th}$ layer of Mamba (24 in total). (a) The approximated attention maps of all image tokens: The row and column axis represent the query and key token index respectively. Brighter color indicates larger attention values. (b) The gradient maps of each image token input after back-propagation: Redder colors indicate larger gradient norms.

## A.2    Resolution Transfer

Table 6: Resolution transfer results on ImageNet dataset.

|                       | Resolution 224 | Resolution 336 |
| --------------------- | -------------- | -------------- |
| DeiT-Base             | 81.8           | 81.6           |
| De-focus Mamba-Base   | 82.0           | 81.6           |

Our model is trained on images with 224 resolution. We test the transfer performance under resolution 336 and report the results in Table 6. The results demonstrate that our De-focus Networks can also transfer to different resolutions effectively.

# B    More Implementation Details

## B.1    Visualization

This subsection discusses the detailed implementation of different visualization methods adopted in our paper, including receptive fields (Fig. 3(a)-(b)), attention maps (Fig. 1(a), Fig. 4(a), Fig. 5(a), Fig. 6(a)), and gradient maps (Fig. 1(b), Fig. 3(c)-(d), Fig. 4(b), Fig. 5(b), Fig. 6(b)).

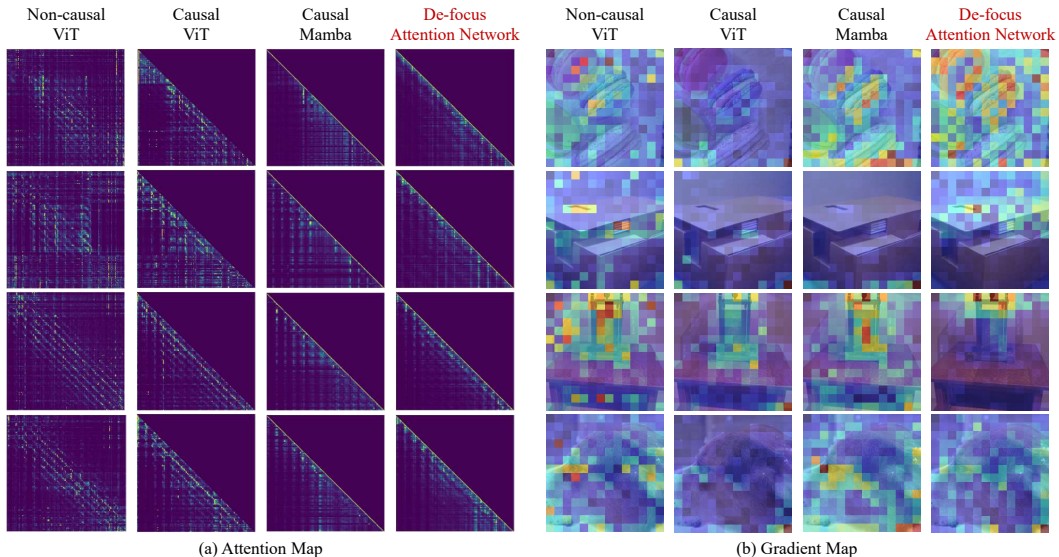

(a) Attention Map      (b) Gradient Map

Figure 5: **Visualizations of Attention Map (a) and Gradient Map (b) of different models**, including non-causal ViT, causal ViT, Causal Mamba and our De-focus Attention Network (Mamba-based). The results are from the 6th layer of ViT (12 in total) and 12th layer of Mamba (24 in total). (a) The approximated attention maps of all image tokens: The row and column axis represent the query and key token index respectively. Brighter color indicates larger attention values. (b) The gradient maps of each image token input after back-propagation: Redder colors indicate larger gradient norms.

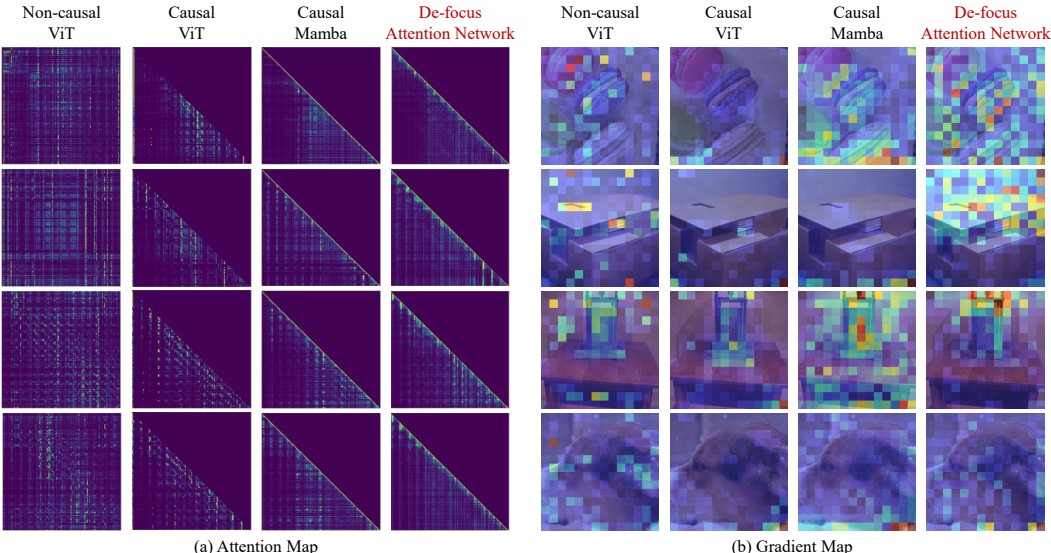

(a) Attention Map      (b) Gradient Map

Figure 6: **Visualizations of Attention Map (a) and Gradient Map (b) of different models**, including non-causal ViT, causal ViT, Causal Mamba and our De-focus Attention Network (Mamba-based). The results are from the 9th layer of ViT (12 in total) and 18th layer of Mamba (24 in total). (a) The approximated attention maps of all image tokens: The row and column axis represent the query and key token index respectively. Brighter color indicates larger attention values. (b) The gradient maps of each image token input after back-propagation: Redder colors indicate larger gradient norms.

**Receptive fields** of a certain layer are defined as the gradient norms of all image tokens on the input side. The gradients here are obtained by back-propagating from the L2-norm of the <CLS> token feature on output side of the same layer. Redder colors indicate larger receptive scores.

**Attention maps**. Similar to receptive fields, the approximated attention maps in our paper are also the gradient norms of all input image tokens (as 'key'). However, different from receptive fields, these gradients come from back-propagation of the feature norm across all image tokens (as 'query') on the same layer's output side. Brighter colors indicate larger attention weights.

**Gradient maps**. Different from receptive fields, the gradient maps of a certain layer are calculated by directly back-propagating from the final training loss to this layer's input image tokens. Then the L2-norm of each image token's gradient is used for plotting the gradient maps. Redder colors indicate larger gradient norms.

By default, the values of receptive fields, attention maps, and gradient maps are divided by the maximum value among all input image tokens for normalization. For attention maps, the diagonal values are set as 0 manually to eliminate the influence induced by residual connection. All image samples are randomly selected.

## B.2   Image Classification

The hyper-parameters for training on ImageNet-1k [13] from scratch are provided in Tab. 7.

The hyper-parameters for pre-training on ImageNet-21k [55] are provided in Tab. 9. The hyper-parameters for finetuning on ImageNet-1k [13] after pre-training are provided in Tab. 10.

## B.3   Object Detection

The hyper-parameters for training on COCO object detection [36] are provided in Tab. 8.

Since an ImageNet-1K pre-trained model is used, to reduce the discrepancy between the resolutions of images in the COCO dataset and those in the ImageNet-1K dataset, several moditifications are made. Spatial decay parameters (e.g., $\Delta$) and position embedding indices are initially scaled down by factors of approximately 4 and 20 (which are COCO resolution to ImageNet-1K resolution ratios), respectively. The order of image tokens is rearranged as shown in Fig.7, with each $224 \times 224$ section of the image first being spanned, followed by concatenation of these spanned sequences in a z-scan order.

## B.4   Contrastive Language-Image Pre-training (CLIP)

The hyper-parameters for Contrastive Language-Image Pre-training on Laion-400m [56] are provided in Tab. 11.

# C   Licenses of Datasets

**ImageNet-1k** [13] is subject to the ImageNet terms of use [65].

**COCO** [36] is subject to the Flickr terms of use [19].

**Laion-400m** [56] is subject to the Laion-400m LICENSE of use [43].

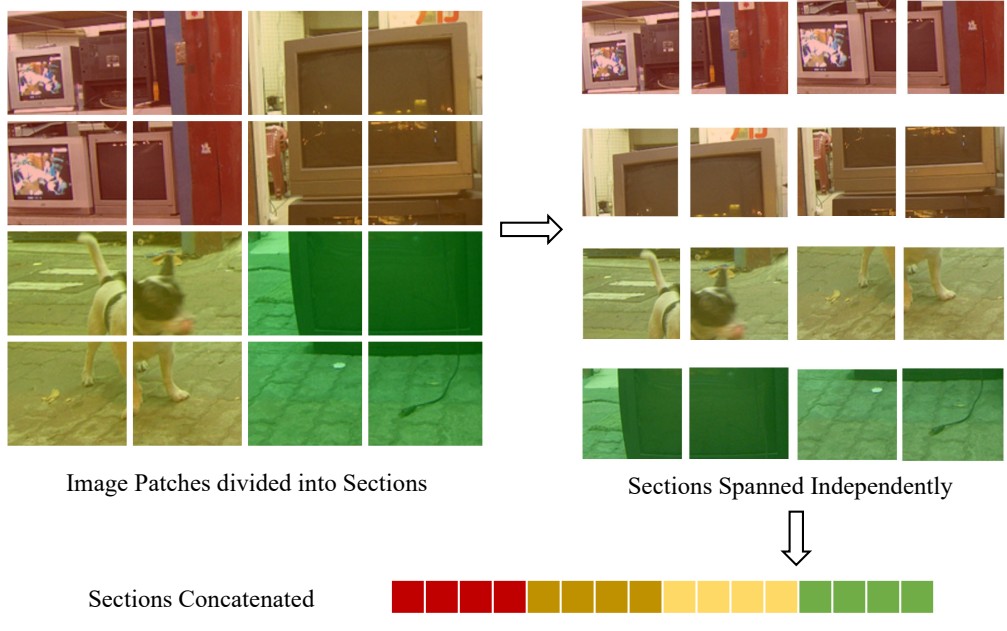

Figure 7: **Rearranging Procedure in Object Detection**. This illustration presents a image divided into several 2x2 sections.

Table 7: **Hyper-parameters for training from scratch on ImageNet-1k.**

| Hyper-parameters | Value |
| --- | --- |
| Input resolution | $224 \times 224$ |
| Training epochs | 300 |
| Warmup epochs | 20 |
| Batch size | 1024 |
| Optimizer | AdamW |
| Peak learning rate | $1.0 \times 10^{-3}$ |
| Learning rate schedule | cosine |
| Weight decay | 0.05 |
| AdamW $\beta$ | (0.9, 0.999) |
| Augmentation | |
|     Color jitter | 0.4 |
|     Rand augment | 9/0.5 |
|     Erasing prob. | 0.25 |
|     Mixup prob. | 0.8 |
|     Cutmix prob. | 1.0 |
|     Label smoothing | 0.1 |
|     repeated augmentation | True |
|     Drop path rate | linear(0.1, 0.7) |

Table 8: **Hyper-parameters for COCO object detection.**

| Hyper-parameters | Value |
|---|---|
| Input resolution | $1024 \times 1024$ |
| Finetuning epochs | 12 / 36 |
| Batch size | 16 |
| Optimizer | AdamW |
| Peak learning rate | $2 \times 10^{-4}$ |
| Learning rate schedule | Step(11) / Step(27,33) |
| Weight decay | $1 \times 10^{-4}$ |
| Adam $\beta$ | (0.9, 0.999) |
| Augmentation | |
|    Random flip | 0.5 |
|    Drop path rate | 0.5 |

Table 9: **Hyper-parameters for pre-training on ImageNet-21k.**

| Hyper-parameters | Value |
|---|---|
| Input resolution | $192 \times 192$ |
| Training epochs | 90 |
| Warmup epochs | 5 |
| Batch size | 4096 |
| Optimizer | AdamW |
| Peak learning rate | $1.0 \times 10^{-3}$ |
| Learning rate schedule | cosine |
| Weight decay | 0.05 |
| AdamW $\beta$ | (0.9, 0.999) |
| Augmentation | |
|    Mixup prob. | 0.8 |
|    Cutmix prob. | 1.0 |
|    Label smoothing | 0.1 |
|    Drop path rate | linear(0.1, 0.5) |

Table 10: **Hyper-parameters for finetuning on ImageNet-1k.**

| Hyper-parameters | Value |
|---|---|
| Input resolution | $384 \times 384$ |
| Finetuning epochs | 20 |
| Warmup epochs | 2 |
| Batch size | 1024 |
| Optimizer | AdamW |
| Peak learning rate | $4 \times 10^{-5}$ |
| Learning rate schedule | cosine |
| Weight decay | 0.05 |
| Adam $\beta$ | (0.9, 0.999) |
| Augmentation | |
|    Mixup prob. | 0.8 |
|    Cutmix prob. | 1.0 |
|    Label smoothing | 0.1 |
|    Drop path rate | linear(0.1, 0.5) |

Table 11: **Hyper-parameters for contrastive vision-language pre-training on Laion-400m.**

| Hyper-parameters | Value |
|---|---|
| Input resolution | $224 \times 224$ |
| Training epochs | 32 |
| Warmup epochs | 20000 iters |
| Batch size | 32768 |
| Optimizer | AdamW |
| Peak learning rate | $5 \times 10^{-4}$ |
| Learning rate schedule | cosine |
| Weight decay | 0.1 |
| AdamW $\beta$ | (0.9, 0.98) |

