# OpenReview forum: "Learning 1D Causal Visual Representation with De-focus Attention Networks"
_NeurIPS.cc/2024/Conference — NeurIPS 2024 poster_

### Official Review · Reviewer_WAfJ · 2024-06-30

**Soundness:** 4
**Presentation:** 4
**Contribution:** 3
**Rating:** 7
**Confidence:** 4

**Summary:**

This paper mainly improves the existing 1D causal models when handling visual inputs, generally 2D non-causal. The key insight of the authors mainly comes from Figure 1: the 1D causal models will over-focus to a few tokens instead of capturing the rich information from the whole image. To fix this issue, the authors propose the de-focus network. They first view the spatial decay and positional embedding as bandpass filters and learn that such filters lead to the capture of diverse information. In addition, the improvement in optimization, especially a significantly larger dropout, also encourages the model to make use of diverse sources of information. Finally, the authors experiment on image classification, object detection, and image-text training to prove the effectiveness of their approach.

**Strengths:**

1. In Figure 1 and Sec. 4.1, the authors make a very good demonstration and reasoning of their intuitions.
2. The De-focus network is simple and reasonable, including both the bandpass filter part and the optimization improvement.
3. The experiments across a wide range of fundamental tasks prove the effectiveness of the De-focus network.

**Weaknesses:**

I think the paper is good overall. I only have one question regarding the generalization of the method, which is not discussed in the paper.

Transformer-based architectures like ViT in image classification and CLIP show certain generalizations to different resolutions. For example, people can train a model on 224x224 and inference on 336x336 by interpolating the positional embeddings. I am curious how well the de-focus network will perform under such generalization scenarios, compared with the 2D non-causal transformer-based architectures.

**Questions:**

See weakness.

**Limitations:**

Yes, the authors have addressed such limitations.

---

> ### Author Rebuttal · Authors · 2024-08-07
>
> We sincerely appreciate Reviewer WAfj for the constructive suggestions. Please see our detailed response below.
>
> ---
>
> > **Q1: Transformer-based architectures like ViT in image classification and CLIP show certain generalizations to different resolutions. For example, people can train a model on 224x224 and inference on 336x336 by interpolating the positional embeddings. I am curious how well the de-focus network will perform under such generalization scenarios, compared with the 2D non-causal transformer-based architectures.**
>
> A1: Thanks for your suggestion. We test the transfer performance under resolution 336 and report the results in the table below.
>
> |   | Resolution 224 (same as training) | Resolution 336 |
> |-------|-------|-------|
> | DeiT-Base | 81.8 | 81.6 |
> | De-focus Mamba-Base | 82.0 | 81.6 |
>
> The results demonstrate that our De-focus Networks can also transfer to different resolutions effectively.

---

> ### Comment · Reviewer_WAfJ · 2024-08-12
>
> Thanks to the authors for the rebuttal! I have also checked the other reviewers' questions and the author's rebuttal. I maintain my current scores for now.

---

### Official Review · Reviewer_J7C5 · 2024-07-01

**Soundness:** 3
**Presentation:** 4
**Contribution:** 3
**Rating:** 4
**Confidence:** 4

**Summary:**

This paper addresses the issue of over-focus in vision models by proposing strategies of using large and scheduled drop path rates and an auxiliary loss on globally pooled features. These strategies aim to encourage the model to focus on a wider range of tokens and improve network optimization. The paper is logically structured, the methods are novel, and the experimental results validate the effectiveness of the proposed approach.

**Strengths:**

1. The paper is well-organized and easy to follow.
2. The authors have an insightful perspective on existing vision models.
3. The research perspective of this paper is interesting and will be inspiring for future studies.

**Weaknesses:**

I am confused about some of the author's viewpoints. For example, in the introduction, the author mentioned that the causal modeling methods explored in existing vision models give them causal reasoning capabilities.

1. Why do vision models based on standard Transformers without adding a causal attention mask still possess some level of causal reasoning abilities?

2. Can we consider that vision Transformers inherently have causal abilities, and that existing improved methods simply amplify these models' causal capabilities?

**Questions:**

1. Can using large spatial dropout/dropblock or randomly dropping masked image tokens achieve similar effects compared to using a Large Drop Path Rate? If these methods are feasible, could additional experimental results comparisons be included?

2. The authors need to provide further explanation on the rationale behind the positioning of learnable decay and learnable RoPE in the attention mechanism. Besides, can we directly deploy them on the feature maps before the attention mechanism?

**Limitations:**

1. The authors compared the parameter counts between models in terms of efficiency, but parameter count alone may not fully reflect model efficiency. I suggest the authors consider adding metrics related to model FLOPs and latency in Table 1 and Table 2.

2.  Adding a classification loss to the model is not a novel method.

3. The ablation study lacks experimental results comparing with other causal methods.

4. There is a small grammatical error. In the paragraph on line 38, the phrase "On one hand xxx, On the other hand xxx" typically indicates a contrasting situation, but in this context, the authors may not intend to convey such a situation.

---

> ### Author Rebuttal · Authors · 2024-08-07
>
> We thank Reviewer J7C5's feedback. It should first be clarified that the reviewer seems to have confused the concept of causal reasoning with 1D causal modeling in our paper. This may have led to some misunderstandings, such as the mistaken belief that we did not compare with other causal modeling methods in the experiments. And other concerns, like additional experiments with dropout, do not constitute valid reasons for rejection. Detailed responses are provided below.
>
> ---
>
> > **Q1: Why do vision models based on standard Transformers without adding a causal attention mask still possess some level of causal reasoning abilities? Can we consider that vision Transformers inherently have causal abilities, and that existing improved methods simply amplify these models' causal capabilities?**
>
> A1: Our paper does not involve the concept of causal reasoning. There is a misunderstanding: causal modeling [1,2] does not refer to causal reasoning, it means that image patches are treated as time series like language, and the patches at the front of the sequence are used to predict subsequent patches. Whether current vision models possess causal reasoning abilities is orthogonal to our paper. We will add the definition of causal modeling in the introduction.
>
> ---
>
> > **Q2: Can using large spatial dropout/dropblock or randomly dropping masked image tokens achieve similar effects compared to using a Large Drop Path Rate? If these methods are feasible, could additional experimental results comparisons be included?**
>
> A2: Thanks for your suggestion. It should be first noted that while these attempts are interesting, they do not diminish the contributions of our paper. We have conducted additional experiments and the results are provided in the table below. It's shown that using large spatial dropout or dropping masked image tokens is inferior to using a large drop path rate.
>
> |   | ImageNet Top1 Acc |
> |-------|-------|
> | Large Drop Path Rate | 82.0 |
> | Large Spatial Dropout | 80.9 |
> | Drop Masked Image Tokens | 80.2 |
>
> ---
>
> > **Q3: (a) The authors need to provide further explanation on the rationale behind the positioning of learnable decay and learnable RoPE in the attention mechanism. (b) Besides, can we directly deploy them on the feature maps before the attention mechanism?**
>
> A3: (a) Please refer to L38-L42 in the introduction and L127-L130 in Section 4.1 for the reason of using learnable decay and RoPE in the attention mechanism. Specifically, the motivation stems from our observation of the "over-focus" issue. Learnable decay helps increase or decrease the network's emphasis on distant data. Learnable RoPE allows the network to focus on different aspects of data within various spatial frequencies. By employing multiple settings for both decay and RoPE, the network can create a more balanced and varied pattern of attention.
>
> (b) Learnable decay and learnable RoPE can not be applied to the feature maps, because the relative positions of different tokens should be taken into account.
>
> ---
>
> > **Q4: The authors compared the parameter counts between models in terms of efficiency, but parameter count alone may not fully reflect model efficiency. I suggest the authors consider adding metrics related to model FLOPs and latency in Table 1 and Table 2.**
>
> A4: It should be noted that efficiency is not the focus of our paper. As in many other papers, parameter counts are reported to ensure a fair comparison between models of similar size, which is not directly related to efficiency. To address the concern, we test the FLOPs and latency of our model, and our "de-focus attention" strategy adds less than 1% to both the model's FLOPs and latency.
>
> ---
>
> > **Q5: Adding a classification loss to the model is not a novel method.**
>
> A5: The novelty of our paper includes the observation of the "over-focus" issue in training causal models, the proposal of the "de-focus attention" strategy, as well as the demonstration that 1D causal models can deliver comparable performances to non-causal models (as noted by the summary of Reviewer WFuU, Reviewer 4dxw and Reviewer WAfJ). It should be emphasized that the use of auxiliary loss is only a part of our overall "de-focus attention" strategy.
>
> ---
>
> > **Q6: The ablation study lacks experimental results comparing with other causal methods.**
>
> A6: We have compared with other causal methods in the main results, including RetNet[3], Mamba[4], Mamba-ND[5]. Please refer to Table 1 in the paper.
>
> ---
>
> > **Q7: There is a small grammatical error. In the paragraph on line 38, the phrase "On one hand xxx, On the other hand xxx" typically indicates a contrasting situation, but in this context, the authors may not intend to convey such a situation.**
>
> A7: Thanks for your advice, we shall fix it in the final version.
>
> ---
>
> [1] Yi Tay, Mostafa Dehghani, Vinh Q. Tran, Xavier Garcia, Jason Wei, Xuezhi Wang, Hyung Won Chung et al. UL2: Unifying Language Learning Paradigms. In The Eleventh International Conference on Learning Representations, 2023.
>
> [2] Erik Nijkamp, Hiroaki Hayashi, Caiming Xiong, Silvio Savarese, and Yingbo Zhou. Codegen2: Lessons for training llms on programming and natural languages. arXiv preprint arXiv:2305.02309, 2023.
>
> [3] Yutao Sun, Li Dong, Shaohan Huang, Shuming Ma, Yuqing Xia, Jilong Xue, Jianyong Wang, and Furu Wei. Retentive network: A successor to transformer for large language models. arXiv preprint arXiv:2307.08621, 2023.
>
> [4] Albert Gu, and Tri Dao. Mamba: Linear-time sequence modeling with selective state spaces. arXiv preprint arXiv:2312.00752, 2023.
>
> [5] Shufan Li, Harkanwar Singh, and Aditya Grover. Mamba-nd: Selective state space modeling for multi-dimensional data. arXiv preprint arXiv:2402.05892, 2024.

---

> > ### Comment · Reviewer_J7C5 · 2024-08-12
> > **Response to authors**
> >
> > I have thoroughly read the authors' responses and revisited the paper. Thank you for the detailed answers. However, I still have some concerns regarding Q1 and Q2.
> >
> > 1. For Transformer models that do not include adding a causal attention mask, how do they possess the ability for causal modeling? Furthermore, based on the current definition of "causal modeling," can we consider CNNs as a form of causal modeling as well?
> >
> > 2. Building on the last question, if existing models already inherently possess causal modeling capabilities, would the proposed method be considered merely an enhancement?
> >
> > 3. The operational mechanism of the Large Drop Path Rate seems similar to large spatial dropout/dropblock or randomly dropping masked image tokens. Why is it that the Large Drop Path Rate can achieve more advantageous results? What are the reasons that lead to its superior performance? Observation from the table in your reply.
> >
> > Many thanks.

---

> > > ### Comment · Reviewer_J7C5 · 2024-08-12
> > > **Regarding the response to Q5**
> > >
> > > Yes, I understand. However, you listed adding an auxiliary classification loss  as one of your main contributions to optimize the model's representation.

---

> > > > ### Author Response · Authors · 2024-08-13
> > > >
> > > > Thank you for your feedback. We understand your concern regarding the novelty of incorporating an auxiliary classification loss.
> > > >
> > > > It's important to emphasize that the auxiliary loss is not a standalone contribution but a component of our comprehensive "de-focus attention" strategy. The core novelty of our work lies in the holistic application of this strategy, which, as demonstrated in our results, enables 1D causal models to achieve performance comparable to non-causal models.
> > > >
> > > > We respectfully suggest that the value of the auxiliary loss should be viewed in the context of its role within our broader strategy, which collectively advances the state of the art in causal modeling.

---

> ### Author Response · Authors · 2024-08-13
>
> Thanks for your reply! We appreciate your time and efforts to read our responses and paper. Please see our new response below.
>
>
> ---
>
> > **Q1: For Transformer models that do not include adding a causal attention mask, how do they possess the ability for causal modeling? Furthermore, based on the current definition of "causal modeling," can we consider CNNs as a form of causal modeling as well?**
>
> A1: It should be first clarified that standard Transformers or CNNs can not learn to perform causal modeling  without a causal attention mask. In mathematical terms, given an input token sequence $x_1, ..., x_n$, causal modeling aims to maximize the likelihood of the correct next token given previous tokens in the sequence, which can be formulated as
> $$\max_\theta\sum_{i=1}^{n-1}\log p(x_{i+1}|x_1, ..., x_i;\theta),$$
> where $\theta$ refers to the parameter set of the model.
>
> For Transformers and CNNs, a causal mask is necessary to restrict access to future tokens. Without a causal mask, Transformers can access the entire input sequence, i.e., they will use $x_1, ..., x_n$ as the probability condition rather than $x_1, ..., x_i$. Similarly, CNNs can access neighborhood tokens in the local window, i.e., they will use $x_{i-w}, ..., x_{i+w}$ as the probability condition, where $w$ is the local window width. These conditions both include the next token, which leads to information leakage. Therefore, standard Transformers and CNNs will rely on such shortcuts, failing to learn to perform causal modeling effectively.
>
>
> ---
>
> > **Q2: Building on the last question, if existing models already inherently possess causal modeling capabilities, would the proposed method be considered merely an enhancement?**
>
> A2: Based on the reasons mentioned above, existing models do not inherently have causal modeling abilities, and our De-focus Attention Networks demonstrate that causal models equipped with the "de-focus attention" strategy can achieve comparable performances with non-causal models.
>
>
> ---
>
> > **Q3: The operational mechanism of the Large Drop Path Rate seems similar to large spatial dropout/dropblock or randomly dropping masked image tokens. Why is it that the Large Drop Path Rate can achieve more advantageous results? What are the reasons that lead to its superior performance? Observation from the table in your reply.**
>
> A3: Large spatial dropout or dropping masked image tokens could theoretically introduce some regularization effects. However, there are nuanced differences in their mechanisms and impacts on model training compared to large drop path rate.
> - Large spatial dropout only drops portions of the tokens, so some tokens can still access information from neighboring tokens. Therefore, the network can still learn to focus on neighboring tokens and rely on depth to gather information.
> - Dropping masked image tokens completely removes some input tokens, so the model may fail to learn to pay attention to tokens that are distant in positions. As a result, it does not prevent the model from leveraging depth to learn representations. Its regularization effect is thus not significant.
>
>
> ---
>
> We hope that our response can address your concerns, and any further discussion is welcomed.

---

### Official Review · Reviewer_4dxw · 2024-07-13

**Soundness:** 3
**Presentation:** 3
**Contribution:** 3
**Rating:** 6
**Confidence:** 4

**Summary:**

The paper addresses the challenges of using 1D causal modeling for images, which traditionally require 2D non-causal modeling due to inherent modality differences between vision and language models. It identifies a significant issue in existing 1D causal vision models termed "over-focus," where the model's attention is overly concentrated on a few visual tokens, hindering diverse feature extraction and optimization. To combat this, the paper introduces De-focus Attention Networks that use learnable bandpass filters to diversify attention patterns and incorporate strategies like large drop path rates and an auxiliary loss on globally pooled features to broaden token attention and enhance model optimization. Experiments on several image underst show that these innovations enable 1D causal visual representation to achieve performance comparable to 2D non-causal models in various tasks, including global perception and multi-modal understanding.

**Strengths:**

+ Identifying over-focus issue in 1D causal visual modeling is an interesting and provide a starting point for the work in thissubmission.

+ A "de-focus attention" strategy is introducted to address this issue. It incorporates learnable bandpass filters into the existing attention mechanisms to generate diverse attention patterns. The proposed idea is resonable.  A high drop path probability and an auxiliary loss is also proposed for better optimization.

+ Experiments on seval visual understanding benchmarks show that via the proposed De-focus Attention Networks, 1D causal visual representation can match the performance of 2D non-causal representation in tasks involving global perception, dense prediction, and multi-modal understanding.

**Weaknesses:**

- For CLIP experiments, it is helpful to also report comparisons for cross-modal retrival on MSCOCO to follow existing routines of CLIP model comparisons.

- The major motivation is about addressing challenges in constructing unified multi-modal model, while the main experments are about image understanding tasks. To this point, seems discussions/explorations about impact of proposed method on MLLM is missing. I'm also curious about its potential applications for image generation models.

**Questions:**

Please refer to details in Weakness part above.

**Limitations:**

Some limitations of the work have been discussed in the submission.

---

> ### Author Rebuttal · Authors · 2024-08-07
>
> We thank Reviewer 4dxw for the thoughtful review and the insights provided. We appreciate the opportunity to discuss the enhancements and implications of our research further.
>
> ---
>
> > **Q1: For CLIP experiments, it is helpful to also report comparisons for cross-modal retrival on MSCOCO to follow existing routines of CLIP model comparisons.**
>
> A1:  In response to your comments, we have conducted additional experiments using the CLIP model on the MSCOCO dataset. These new results further reinforce the findings presented in our initial submission, emphasizing the robustness and applicability of our approach across different data sets.
>
> | Model | Causal | Image Retrieval Recall@1 | Image Retrieval Recall@5 | Image Retrieval Recall@10 |  Text Retrieval Recall@1 |  Text Retrieval Recall@5 |  Text Retrieval Recall@10 |
> |-------|-------|-------|-------|-------|-------|-------|-------|
> | OpenAI CLIP-Base/32 | No | 30.4 | 55.0 | 65.7 | 49.2 | 73.4 | 82.4 |
> | OpenCLIP-Base/32 | No | 35.3 | 61.0 | 71.8 | 52.5 | 77.0 | 84.9 |
> | De-focus Mamba-Base/32 | Yes | 34.6 | 60.3 | 71.2 | 51.7 | 76.3 | 84.8 |
>
> ---
>
> > **Q2: The major motivation is about addressing challenges in constructing unified multi-modal model, while the main experiments are about image understanding tasks. To this point, seems discussions/explorations about impact of proposed method on MLLM is missing.**
>
> A2: Thank you for your suggestions. In our research, we mainly study the application of causal modeling to images and achieve competitive performance. These results show a promising path towards a unified framework for joint causal modeling of images and text.
>
> However, the integration of these modalities remains a considerable challenge. Models such as Fuyu-8B[1] and Chameleon[2] that attempt to fuse image and text modeling are examples in this regard. Unfortunately, the details of these models and their training process are not publicly available, and they have not yet achieved the performance level of state-of-the-art MLLMs. This limitation underscores the crucial need for ongoing research in this domain to bridge the existing gap.
>
> ---
>
> > **Q3: I'm also curious about its potential applications for image generation models.**
>
> A3:  Recent preliminary studies, such as "Autoregressive Model Beats Diffusion: Llama for Scalable Image Generation"[3], have begun to uncover the potential of causal modeling in image generation. While our work primarily focuses on perception and understanding, these two works together indicate that causal modeling is a viable alternative to traditional methods across various areas of AI research.
>
> ---
>
> [1] Rohan Bavishi, Erich Elsen, Curtis Hawthorne, Maxwell Nye, Augustus Odena, Arushi Somani, and Sağnak Taşırlar. Introducing our multimodal models, 2023.
>
> [2] Chameleon Team. Chameleon: Mixed-modal early-fusion foundation models. arXiv preprint arXiv:2405.09818, 2024.
>
> [3] Sun, Peize, et al. Autoregressive Model Beats Diffusion: Llama for Scalable Image Generation. arXiv preprint arXiv:2406.06525, 2024.

---

### Official Review · Reviewer_WFuU · 2024-07-15

**Soundness:** 2
**Presentation:** 3
**Contribution:** 3
**Rating:** 4
**Confidence:** 2

**Summary:**

The paper explores the feasibility of representing images with 1D causal modeling in unified multi-modal vision and language models, addressing the "over-focus" issue in existing models by proposing De-focus Attention Networks (DANs) with learnable bandpass filters and enhanced training strategies. Extensive experiments show that 1D causal visual representation can perform comparably to 2D non-causal representation in global perception, dense prediction, and multi-modal understanding tasks, with code to be released.

**Strengths:**

1. The proposal to use 1D causal modeling for images is a novel approach that challenges the traditional 2D non-causal representation, opening new avenues for unified multi-modal models.

2. The use of large and scheduled drop path rates and an auxiliary loss on globally pooled features are effective training strategies that promote broader attention and better network optimization.

**Weaknesses:**

1. The evaluation of the proposed method is inconsistent, e.g., in table 1, the performance of Base ViT model is reported, but Small and Large model are missing. In table 2 and 3, only Mamba model results are compared.

**Questions:**

Please refer to the weakness section.

**Limitations:**

Limitations have been discussed in this paper.

---

> ### Author Rebuttal · Authors · 2024-08-07
>
> We appreciate Reviewer WFuU for the comments, yet we must emphasize that the weakness Reviewer WFuU has identified is not a sufficient reason for rejection. Our experiments based on Mamba have demonstrated that 1D visual causal modeling can achieve comparable performance to non-causal models. Other results based on ViT are included primarily to supplement this evidence. Their performance, whether superior or not, does not compromise the soundness of the arguments or the validity of the conclusions drawn.
>
> ---
>
> > **Q1: The evaluation of the proposed method is inconsistent, e.g., in table 1, the performance of Base ViT model is reported, but Small and Large model are missing. In table 2 and 3, only Mamba model results are compared.**
>
> A1: The aim of our research is to assess the feasibility and performance potential of causal modeling, a goal we have demonstrated by using the Mamba-based model. The structure of specific models, such as the ViT-based model mentioned in the paper, serves merely to provide additional evidence, further proving that our method works well and can be applied generally.
>
> Due to computational constraints, we conducted only the most essential experiments and presented these in the paper. Addressing your concerns, we have also conducted supplementary experiments employing the ViT model:
>
> - We began by training the De-focus ViT-Small model on the ImageNet dataset. The results of this experiment are depicted in the following table.
>
> |  | Causal | ImageNet Top1 Acc |
> |-----|----|----|
> | (DeiT) ViT-Small | No | 79.9 |
> | De-focus ViT-Small | Yes | 79.6 |
> | De-focus Mamba-Small | Yes | 80.3 |
>
>
> - Additionally, we conducted an experiment in a detection task setting. We did not change the existing detection architecture but instead utilized the ViT-base model listed in Table 1 as the pretrained model.
>
> |  | Causal | Epoch | AP | AP50 | AP75 |
> |-----|----|----|----|----|----|
> | (DeiT) ViT-Base | No | 12 | 49.1 | 69.9 | 52.7 |
> | De-focus ViT-Base | Yes | 12 |  48.9 | 67.1 | 53.3 |
> | De-focus Mamba-Base | Yes | 12 |  50.8 | 68.9 | 55.2 |
>
>
> These additional experiments highlight the adaptability of our approach. Nonetheless, the absence of these experiments from the initial submission does not weaken our main points or lessen the reliability of the conclusions we made.
>
> ---

---

### Decision · Program_Chairs · 2024-09-25

**Decision:**

Accept (poster)

**Comment:**

Paper was reviewed by four expert reviewers and post-rebuttal received somewhat divergent ratings of: 2 x Borderline reject, 1 x Weak Accept, 1 x Accept. Overall, most reviewers agree that the proposed 1D causal attention is interesting and is a viable alternative to more traditional attention mechanisms. However, some concerns have also been raised by the reviewers, including with (1) inconsistent evaluation [WFuU], (2) effectiveness of a large drop path rate compared to large spatial dropout/dropblock [J7C5], (3) ability to generalize to different image resolutions [WAfJ], as well as a couple of others. In general, authors have addressed most of these comments in the rebuttal, including with additional experimental evidence.

Unfortunately, [WFuU] has not engaged in the discussion; [J7C5] has engaged in the discussion and acknowledged that some of the concerns have been alleviated in author responses, but remained concerned with novelty of auxiliary loss. The remaining two reviewers appear happy with author responses and argue for acceptance. Overall, all reviews agree that "Contributions" of the work are "good" and most feel that "Presentation" is "excellent". Only [WFuU] raised issue with "Soundness", rating it "fair", while other reviewers rated it "good" or "excellent".

AC had considered the reviews, rebuttal, discussion that followed and the paper itself. AC agrees that the work is interesting and may be broadly valuable (e.g., the learnable band pass filter can be broadly useful in attention mechanisms). While AC acknowledges that more experiments in additional tasks (e.g., VLMs) would make the work even stronger, this should not take away from contributions already made by the paper. Overall, AC agrees with more favorable reviewers that the work is solid, leading to an Acceptance recomendation.